# Prevalence and Correlates of Sexual Risk Behavior among School-Going Adolescents in Four Caribbean Countries

**DOI:** 10.3390/bs10110166

**Published:** 2020-10-29

**Authors:** Supa Pengpid, Karl Peltzer

**Affiliations:** 1ASEAN Institute for Health Development, Mahidol University, Salaya, Phutthamonthon, Nakhon Pathom 73170, Thailand; supa.pen@mahidol.ac.th; 2Department of Research Administration and Development, University of Limpopo, Turfloop 0727, South Africa; 3Department of Psychology, University of the Free State, Bloemfontein 9300, South Africa

**Keywords:** sexual behavior, adolescents, health risk behavior, Caribbean

## Abstract

This study aimed to assess the prevalence and correlates of sexual risk behaviors among adolescents in the Caribbean. Nationally representative cross-sectional data were analyzed from 9143 adolescents (15 years = median age) that took part in the 2016 Dominican Republic, 2016 Suriname, 2017 Jamaica, and 2017 Trinidad and Tobago Global School-Based Student Health Survey (GSHS). The results indicate that 41.4% of the students had ever had sex, ranging from 26.4% in Trinidad and Tobago to 48.1% in Jamaica. Among the sexually active, 58.8% had had ≥2 sexual partners; 58.6% had had an early sexual debut (≤14 years); 41.9% had not used birth control the last time they had sex; 28.4% had not used a condom the last time they had sex; and, of the whole sample, 31.9% had engaged in two or more (multiple) sexual risk behaviors, ranging from 16.5% in Trinidad and Tobago to 40.3% in Jamaica. In an adjusted logistic regression analysis, substance use (tobacco, alcohol, and cannabis), psychological distress, frequent soft drink intake, participation in physical fighting, school truancy, older age, and male sex were associated with single and/or multiple sexual risk behaviors. A large number of adolescents in the Caribbean reported sexual risk behaviors, emphasizing the need for intervention.

## 1. Background

Sexual initiation often occurs during the adolescent period and is associated with unprotected sex and other sexual risk behaviors [1]. Among adolescents in Caribbean countries, various sexual risk behaviors, such as early sexual debut and unprotected sex, have been identified, including its consequences of sexually transmitted infection and adolescent pregnancy [2,3]. The adolescent fertility rate (defined as the number of births per 1000 women aged 15 to 19 years) is 63.8 (per 1000 adolescent women) in the Caribbean (108.7 in the Dominican Republic and 77.3 in Jamaica), which surpassed the global average of 55.7 per 1000 adolescent women [2]. Overall, the Caribbean adult (15–49 years) HIV prevalence was 1.1%, and one third of new HIV infections in the Caribbean in 2019 were among young people aged 15–24 [4]. About 40% of boys and 20% of girls (13–15 years) had initiated sex [2], and of those sexually active, 79% of boys and 56% of girls had had an early sexual debut (<14 years) in the Caribbean [2]. Almost one in three boys (31%) and 10% of girls (13–15 years of age) had multiple sexual partners [2], and 38% did not use a condom the last time they had sex [2]. The prevalence of early sexual debut (<15 years) was 26.9% in six Caribbean countries in 2007–2009 (13–16 years) [5]. In several local studies among adolescents in Jamaica, high sexual risk behavior has been reported—e.g., in a subsample of adolescents (15–19 years) in 2008–2009, 32% of females and 54% of males had had sexual initiation, and among sexually active girls and boys 12% and 52%, respectively, had had multiple sexual partners [6]. An adolescent school survey (13–18 years) in Kingston and St Andrew in 2012 showed that 58.6% reported sexual initiation and 31.8% had had multiple sexual partners [7]; among sexually active youth (15–24 years) in Jamaica in 2012, 58.8% reported inconsistent condom use and 44.5% had had multiple sexual partners [8].

In a review among adolescents in the English-speaking Caribbean, “the quality of the parent-adolescent relationship, the presence of violence, substance abuse or mental health problems in the family, peer relationships, cultural attitudes, and a history of physical and sexual abuse” were associated with several sexual and reproductive health outcomes [9]. Globally, as reviewed in Peltzer and Pengpid [10] (p. 406), “factors associated with sexual risk behavior among adolescents (ever had sex, early sexual debut, no condom use, and no contraceptive use), include, male sex, older age, substance use, psychological distress, school truancy, lack of parental and peer support.” In addition, in a multi-country adolescent study, frequent bullying victimization increased the odds of sexual initiation, non-condom use, and multiple sex partners [11]; frequent involvement in physical fighting increased the odds of sexual initiation and multiple sex partners [12]; in American adolescents, frequent soda consumption increased the odds of multiple sexual partners and unprotected sex [13]; and in a study of college students in Zambia, low physical activity increased the odds of any risky sexual behavior [14].

However, there is a lack of recent national data on adolescent sexual risk behavior and its correlates in the Caribbean, such as in Dominican Republic, Jamaica, Suriname, and Trinidad and Tobago. Knowing the occurrence and factors associated with sexual risk behavior and its associated factors among adolescents in Caribbean countries will help in informing intervention strategies targeting the delay of sexual initiation and promoting “safer sex”. According to Joint United Nations Programme on HIV/AIDS (UNAIDS), “Safer sex strategies include postponing sexual debut, non-penetrative sex, correct and consistent use of male or female condoms, and reducing the number of sexual partners” [15] (p. 11).

Therefore, this study intended to estimate the prevalence and predictors of sexual risk behaviors among school adolescents in Dominican Republic, Jamaica, Suriname, and Trinidad and Tobago in 2016–2017.

## 2. Methods

### 2.1. Sample and Procedures

Cross-sectional nationally representative survey data from the 2016 Dominican Republic, 2016 Suriname, 2017 Jamaica, and 2017 Trinidad and Tobago GSHS were analyzed [16]. These countries were selected on the basis of all the countries in the Caribbean where recent GSHS data were available. A two-stage cluster sample design was utilized to generate representative data of all students in Forms 1–6 in the Dominican Republic, Jamaica, Suriname, and Trinidad and Tobago [16]. At the first stage, schools were selected with a probability proportional to the enrolment size, and at the second stage classes were randomly selected and all the students in the selected classes were eligible to participate [16]. Data collection was done with a self-administered multichoice format questionnaire translated into the country language under the supervision of trained external survey administrators in the classroom [16]. Students recorded their responses on computer scannable sheets, and student privacy was maintained through anonymous and voluntary participation [16]. More detailed information can be publicly accessed [16]; the overall response rate was 63% in the Dominican Republic, 60% in Jamaica, 83% in Suriname, and 89% in Trinidad and Tobago [16]. National ethics committees approved the study and written informed consent was obtained from the participating schools, parents, and students [16].

The study context characteristics of the four study countries are described in Table 1.

### 2.2. Measures

The GSHS questionnaire used is shown in Table 2 [16]. The GSHS measure draws on content from the Centers for Disease Control and Prevention (CDC) Youth Risk Behavior Survey, for which the test- and retest reliability has been established [22]. A study examining the test-retest reliability of the GSHS measure among Fijian girls found an “average agreement between test and retest was 77%, and average Cohen’s kappa was 0.47.” [23] (p. 181).

Sexual risk behavior was assessed with questions on ever having had sexual intercourse, the age of sexual debut, the number of people having had sexual intercourse within a lifetime, condom use at the last sexual intercourse, and birth control use at the last sexual intercourse. Individual sexual risk behaviors were defined as ever having had sex, early sexual debut (≤14 years), having had two or more sexual partners in a lifetime, non-condom use in the last sexual encounter, and non-birth control use in the last sexual encounter. A composite sexual risk behavior measure included having had sex, early sexual debut (≤14 years), having had two or more sexual partners in a lifetime, and non-condom use in the last sexual encounter; non-birth control use was excluded due to overlap with non-condom use in the last sexual encounter.

Emotional-contextual factors include factors such as bullying victimization, being in physical fights, hunger, parental tobacco use, passive smoking, school truancy, and psychological distress variables (loneliness, anxiety, no close friends, suicidal ideation, and suicide attempt) that were summed and grouped into 0 = 0, 1 = 1 single, and 2–5 = 2 multiple psychological distress [24].

Health risk behaviors included current tobacco use, current cannabis use, lifetime drunkenness, attendance of physical education, injury, and soft drink consumption.

Protective factor items included peer support and four parental support items (connectedness, supervision, bonding, and respect for privacy). The latter were summed and classified into three groups, 0–1 low, 2 medium, and 3–4 high support [24].

### 2.3. Data Analysis

Secondary data analyses were conducted using “STATA software version 15.0 (Stata Corporation, College Station, TX, USA)”. Descriptive statistics were used to describe the sample. Logistic regression was used on the whole sample to identify the predictors of individual sexual risk behaviors (non-birth control use in the last sexual encounter, non-condom use in the last sexual encounter, multiple sexual partners, early sexual debut, and ever having had sex) and a summary measure of four sexual risk behaviors (having ≥2 sexual risk behaviors). Co-variates were included based on previous literature reviews [11,12,13,14,24]. Taylor linearization procedures were utilized in all the statistical operations to account for the sampling weight and the multistage design of the study. Missing cases were not included in the analysis. The level of significance was set at *p* < 0.05.

## 3. Results

### 3.1. Characteristics of the Sample and Sexual Behavior

The sample consisted of 9143 school adolescents from the Dominican Republic, Jamaica, Suriname, and Trinidad and Tobago; the overall mean age was 15.3 years (SD = 1.6). More than two in five students (41.4%) had ever had sex, ranging from 26.4% in Trinidad and Tobago to 48.1% in Jamaica. Among those who had ever been sexually active, 58.6% had had an early sexual debut (≤14 years), ranging from 47.9% in Trinidad and Tobago to 67.0% in Jamaica; 58.8% had had ≥2 sexual partners, ranging from 43.3% in Trinidad and Tobago to 65.0% in Jamaica; 28.4% had used condoms in their last sexual encounter, ranging from 26.6% in the Dominican Republic to 33.8% in Trinidad and Tobago; 41.9% had used methods of birth control in their last sexual encounter, ranging from 36.0% in the Dominican Republic to 52.4% in Jamaica. Of the whole sample, 31.9% were involved in multiple (≥2) sexual risk behaviors, ranging from 16.5% in Trinidad and Tobago to 40.3% in Jamaica. Additional characteristics of the sample are described in Table 3.

### 3.2. Associations with Sexual Risk Behavior

In an adjusted logistic regression analysis, male sex was associated with ever having had sex (AOR: 4.41, 95% CI: 3.62–5.36), an early sexual debut (AOR: 7.26, 95% CI: 5.89–9.62), multiple sexual partners (AOR: 7.95, 95% CI: 5.39–11.84), non-condom use in the last sexual encounter (AOR: 2.89, 95% CI: 1.91–4.40), non-birth control use in the last sexual encounter (AOR: 3.84, 95% CI: 2.45–6.02), and multiple sexual risk behavior (AOR: 7.16, 95% CI: 5.61–9.15). Compared to participants aged 15 years or less, the participants aged 16 years or older were more likely to have ever had sex (AOR: 2.75, 95% CI: 1.66–4.55), to have had multiple sexual partners (AOR: 2.59, 95% CI: 1.84–3.65), to have not used a condom in their last sexual encounter (AOR: 1.73, 95% CI: 1.31–2.28), to have not used birth control in their last sexual encounter (AOR: 1.74, 95% CI: 1.14–2.65), and to have engaged in multiple sexual risk behaviors (AOR: 2.89, 95% CI: 1.68–4.98).

Regarding emotional-contextual factors, students with high psychological distress had increased odds of sexual initiation, non-condom use, multiple sexual partners, early sexual debut, and multiple sexual risk behaviors. Having been frequently involved in physical fights was associated with multiple sexual risk behaviors, sexual initiation, early sexual debut, non-birth control use, and multiple sexual partners. Parental tobacco use increased the odds for having ever had sex and non-birth control use, while school truancy was positively associated with ever having had sex and multiple sexual risk behaviors. Bullying victimization, hunger, and passive smoking were not significantly associated with any of the sexual risk behaviors.

In terms of health risk behaviors, current tobacco use was associated with an early sexual debut, multiple sexual partners, and multiple sexual risk behaviors, while current cannabis use and ever having been drunk were associated with four sexual risk behaviors. Frequent soft drink intake increased the odds of non-birth control use, multiple sexual partners, and multiple sexual risk behaviors. Nonattendance of physical education and frequent physical injury were not significantly associated with any of the sexual risk behaviors.

Regarding protective factors, high peer support increased the odds of multiple sexual partners, and moderate parental support was protective against non-condom use (see Table 4 and Table 5).

## 4. Discussion

The results show a high proportion of sexual initiation (41.4%) among school-going adolescents in four Caribbean countries, which seems higher than the 2013 estimates for the Caribbean (20% among girls and 40% among boys) [2], in a community survey among adolescents in seven African countries (25.9%) [25]; among school adolescents in 30 countries in Europe, Israel, and Canada (27%) [26]; and in a 10 European countries study among adolescents (18.8%) [27]. There was some country variation in the prevalence of sexual initiation, with Jamaica being the highest (48.1%) and Trinidad and Tobago being the lowest (26.4%). Two previous local studies among adolescents in Jamaica also reported a high proportion of sexual initiation (32% among girls and 54% among boys) [6], and in Kingston and St Andrew (58.6%) [7].

Among sexually active students, 28.4% had not used condoms in their last sexual encounter, which is a significant improvement compared to 2013 or earlier data among adolescents in the Caribbean (38%) [2], Haiti (42.3%) [3], seven African countries (46% among girls and 49% among boys) [25], and four countries in Southeast Asia (46.9%) [24], but similar to adolescents in Uganda (22.7%) [28]. Moreover, high sexual risk behavior was found in terms of early sexual debut (≤15 years) (58.8%), multiple sex partners (58.8%), non-birth control use (last sex) (41.9%), and engagement in two or more sexual risk behaviors (31.9%). The proportion of early sexual debut among adolescents in the four Caribbean countries was higher in this study (58.8%) than in a previous study in the Caribbean in 2007–2009 (26.9%) [5] and in seven African countries (<15 years, 21% among girls and 28% among boys) [18], but similar to 2013 or earlier Caribbean research data (56% of girls and 79% of boys) [2]. The proportion of multiple sexual partners (58.8%) among adolescents in the Caribbean in this study was higher than in 2013 research in the Caribbean (31% among boys and 10% among girls) [2], in Kingston and St Andrew in Jamaica (31.8%) [7], in another study in Jamaica (12% among girls and 52% among boys) [6], in Ghana (32.5%) [29], and in 15 year-olds in 10 European countries (52.4%) [27], but lower compared to a local study of 200 adolescents in Haiti (62.2%) [3].

The proportion of non-contraceptive use (41.9%) among adolescents in the Caribbean in this study was higher than in Europe, Israel, and Canada (14%) [26]. Although the prevalence of birth control use (64%) and condom use (73.4%) in the last sexual encounter among adolescents in Dominican Republic in this study was the highest among the four Caribbean study countries, the adolescent fertility rate was the highest in the Caribbean (108.7 per 1000 adolescent women) [2]. This disparity may be related to our sample selection of middle school students rather than out-of-school adolescents. Most sexual risk behaviors were the highest in Jamaica and the lowest in Trinidad and Tobago, which concurs with some previous research [6,7]. Overall, it appears that protected sexual intercourse increased but also early sexual debut and multiple sexual partners increased among adolescents in the four Caribbean countries. It is possible that increased HIV prevention campaigns led to a high proportion of condom use among the studied adolescents.

Male sex and/or older adolescents were associated with almost all individual sexual risk behaviors (sexual initiation, multiple sexual partners, early sexual debut, and unprotected sex) and multiple sexual risk behaviors. Similar results were shown in previous investigations [3,10,24,29,30], which may support the case of sexual risk intervention programs targeting male adolescents at an earlier age than their female counterparts. Unlike in former research [31], this study showed a non-association between frequent hunger experiences (or lower socioeconomic status) and sexual risk behaviors. A possible explanation for this finding is that the prevalence of hunger was low in this study and the concept of socioeconomic status was assessed more comprehensively in other studies, such as including the education of the household head and a list of household possessions [31]. Frequent involvement in physical fights increased the odds for all individual sexual risk behaviors (except for non-condom use) and multiple sexual behaviors. These findings are in line with those of a previous multi-country study among school adolescents [12]. In addition, frequent soft drink consumption was positively associated with non-birth control use, multiple sexual risk behaviors, and multiple sexual partners in this study. In previous research, soft drink intake has been found o be associated with various health risk behaviors, including having had multiple sexual partners and unprotected sex [13,32]. Further research is needed to investigate this novel finding in the Caribbean.

In agreement with a number of previous studies [3,9,29,30,33], this survey showed that substance use (tobacco, cannabis, and alcohol) increased the likelihood of engaging in most individual as well as multiple sexual risk behaviors. In line with previous research [10,29,34,35,36], psychological distress was in this study associated with most single and multiple sexual risk behaviors. The findings from this research seem to show a clustering between psychological distress (an internalizing factor), substance use, interpersonal violence such as physical fighting (an externalizing factor), and sexual risk behavior. This finding may have implications for integrating psychological distress, substance use, and interpersonal violence in sexual and reproductive health promotion programs.

The study showed that attending school was protective against sexual initiation and having multiple sexual risk behaviors, which is consistent with previous research findings [10,30]. Interventions preventing school truancy and promoting school attendance may also be beneficial in sexual risk behavior reduction. Unlike some former research studies [3,9,30,37,38], this survey showed that peer and parental support was not protective against sexual risk behavior. In fact, high peer support increased the odds of having multiple sexual partners. It is possible that this behavior (having multiple sexual partners) is supported by peers and consequently increases it. In a study among school adolescents in the Bahamas, “greater perceived peer risk involvement predicted higher sexual risk behavior index scores” [38]. This could mean that sexual health promotion should include peer norms and perceptions. Comprehensive and timely sexual and reproductive health education is needed in secondary schools in the Caribbean [39].

### Limitations of the Study

The GSHS only includes adolescents that attend school, excluding out-of-school youth. Adolescents who have dropped out of school may be more vulnerable to sexual risk behavior. The GSHS was cross-sectional by design, which precludes causative inference between the study variables. Furthermore, the self-reported data collection may have led to biased responses, in particular regarding sensitive issues such as sexual behavior. The GSHS does not provide a definition of “sexual intercourse”, and therefore it is possible that some students misinterpreted the meaning, but the same question is used in various other surveys among adolescents (e.g., [27]).

## 5. Conclusions

More than two in five students had ever had sex among school-going adolescents in four Caribbean countries. Among the sexually active participants, two in five had had non-birth control use, almost three in five had had an early sexual debut and multiple sexual partners, and almost three in ten had not used condoms in their last sexual encounter. Sexual risk behaviors were higher in students who had psychological distress, engaged in substance use, had frequent soft drink intake, were older, were male, and were absent from school. Taking the identified factors associated with sexual risk behaviors into account will be important in the design and scaling up of sexuality and reproductive health education among school adolescents in the Dominican Republic, Jamaica, Suriname, and Trinidad and Tobago.

### Availability of Data and Materials

The data for the current study are publicly available at the World Health Organization NCD Microdata Repository (URL: https://extranet.who.int/ncdsmicrodata/index.php/catalog).

## Figures and Tables

**Table 1 behavsci-10-00166-t001:** Study context characteristics.

Variable	Dominican Republic [17]	Jamaica [18]	Suriname [19]	Trinidad and Tobago [20]
Population	10,499,704	2,808,570	609,569	1,208,789
Urban population	82.5%	56.3%	66.1%	53.2%
Major ethnic groups	mixed 70.4% (mestizo/indio 58%, mulatto 12.4%), black 15.8%	black 92.1%, mixed 6.1%	Hindustani 27.4%, “Maroon” 21.7%, Creole 15.7%, Javanese 13.7%	East Indian 35.4%, African descent 34.2%, mixed/other 15.3%
Secondary school gross enrolment ratio, female [21]	85%	86%	89%	88%
Secondary school gross enrolment ratio, male [21]	78%	85%	67%	83%
Study sample: Mean age (SD), age range	14.9 (1.5), 11–18 years	15.0 (1.3), 11–18 years	14.8 (1.7), 11–18 years	14.8 (1.7), 11–18 years
School grade				
1st Form	426 (29.4%)	41 (2.5%)	642 (31.0%)	896 (23.3%)
2nd Form	394 (27.2%)	412 (25.0%)	678 (32.7%)	667 (17.4%
3rd Form	317 (21.9%)	557 (33.8%)	691 (33.3%)	851 (22.1%)
4th Form	216 (14.9%)	388 (23.5%)	61 (2.9%)	767 (20.0%)
5th Form	94 (6.5%)	208 (12.6%)	0 (0.0%)	462 (12.0%)
6th Form	0 (0.0%)	44 (2.7%)	0 (0.0)	200 (5.2%)

**Table 2 behavsci-10-00166-t002:** Questionnaire items.

Indicator	Item	Responses (Coding Scheme)
Sex	“What is your sex?”	“Male, Female”
Age	“How old are you?”	“11 years old or younger to 18 years old or older”
Sexual behavior		
Sexual initiation	“Have you ever had sexual intercourse?”	“Yes, No” (coded yes = 1, no = 0)
Age of sexual initiation	“How old were you when you had sexual intercourse for the first time?”	“I have never had sexual intercourse 11 years old or younger to 18 years old or older”
Number of sex partners	“During your life, with how many people have you had sexual intercourse?”	“I have never had sexual intercourse, 1 person to 6 or more people”
Condom use	“The last time you had sexual intercourse, did you or your partner use a condom?”	“I have never had sexual intercourse, Yes, No, I do not know”
Birth control use	“The last time you had sexual intercourse, did you or your partner use any method of birth control, such as withdrawal, rhythm (safe time), birth control pills, or any other method to prevent pregnancy?”	“I have never had sexual intercourse, Yes, No, I do not know”
**Emotional-contextual factors**		
No close friends	“How many close friends do you have?”	“1 = 0 to 4 = 3 or more (coded 1+ = 0, 0 = 1)”
Loneliness	“During the past 12 months, how often have you felt lonely?”	“1 = never to 5 = always (coded 1–3 = 0 and 4–5 = 1)”
Worry/Anxiety	“During the past 12 months, how often have you been so worried about something that you could not sleep at night?”	“1 = never to 5 =a lways (coded 1–3 = 0 and 4–5 = 1)”
Suicidal ideation	“During the past 12 months, did you ever seriously consider attempting suicide?”	“Yes, No”
Suicide attempt	“During the past 12 months, how many times did you actually attempt suicide?”	“1 = 0 times to 5 = 6 or more times (coded 1 = 0 and 2–5 = 1)”
Bullied	“During the past 30 days, on how many days were you bullied?”	“1 = 0 days to 7 =All 30 days (1–2 = 0 and 3–7 = 1)”
Physical fights	“During the past 12 months, how many times were you in a physical fight?”	“1 = 0 times to 8 = 12 or more times (1–2 = 0 and 3–8 = 1)”
Hunger	“During the past 30 days, how often did you go hungry because there was not enough food in your home?”	“1 = never to 5 = always (coded 1–3 = 0 and 4–5 = 1)”
Parental tobacco use	“Which of your parents or guardians use any form of tobacco?”	“1 = neither to 4 = both (coded 1 = 0 and 2–4 = 1)”
Passive smoking	“During the past 7 days, on how many days have people smoked in your presence?”	“1 = 0 days to 5 = all 7 days (coded 1–4 = 0 and 5 = 1)”
School truancy	“During the past 30 days, on how many days did you miss classes or school without permission?”	“1 = 0 days to 10 or more days (coded 1–2 = 0 and 3–5 = 1)”
**Health risk behaviors**		
Current tobacco use	“During the past 30 days, on how many days did you smoke cigarettes/use any tobacco products other than cigarettes, such as country specific examples?”	“1 = 0 days to 7 = All 30 days (coded 1 = 0 and 2–7 = 1)”
Current cannabis use	“During the past 30 days, how many times have you used marijuana?”	“1 = 0 times to 5 = 20 or more times (coded 1 = and 2–5 = 1)”
Ever drunk	”During your life, how many times did you drink so much alcohol that you were really drunk?”	“1 = 0 times to 4 = 10 or more times (coded 1 = 0 and 2–4 = 1)”
Physical education	“During this school year, on how many days did you go to physical education (PE) class each week?”	“1 = 0 days to 6 = 5 or more days (coded 1 = 1 and 2–6 = 0)”
Injury	“During the past 12 months, how many times were you seriously injured?”	“1 = 0 times to 8 = 12 or more times (coded 1–2 = 0 and 3–8 = 1)”
Soft drink intake	“During the past 30 days, how many times per day did you usually drink carbonated soft drinks, such as country specific examples? (Do not include diet soft drinks.)”	“1 = never to = 5 or more times a day (coded 1–3 = 0 and 4–6 = 1)”
**Protective factors**		
Peer support	“During the past 30 days, how often were most of the students in your school kind and helpful?”	“1 =n ever to 5 = always (coded 1–2 =0, 3 = 2 and 4–5 =1)”
Parental supervision	“During the past 30 days, how often did your parents or guardians check to see if your homework was done?”	“1 = never to 5 = always (coded 1–3 = 0 and 4–5 = 1)”
Parental connectedness	“During the past 30 days, how often did your parents or guardians understand your problems and worries?”	“1 = never to 5 = always (coded 1–3 = 0 and 4–5 = 1)”
Parental bonding	“During the past 30 days, how often did your parents or guardians really know what you were doing with your free time?”	“1 = never to 5 = always (coded 1–3 = 0 and 4–5 = 1)”
Parental respect for privacy	“During the past 30 days, how often did your parents or guardians go through your things without your approval?”	“1 = never to 5 = always (coded 1–3 = 0 and 4–5 = 1)”

**Table 3 behavsci-10-00166-t003:** Sample characteristics and sexual behavior types among adolescents in four Caribbean countries, 2016–2017.

Study Variable	All	Ever Sex	Early Sexual Debut ^a^	Multiple Sexual Partners ^a^	Non-Condom Use ^a^	Non-Birth Control Use ^a^	Multiple Risk Factors
	N (%)	N = 2691	N = 1479	N = 1374	N = 794	N = 1074	N = 1888
Sociodemographic variables		%	%	%	%	%	%
All	9143	41.4	58.6	58.8	28.4	41.9	31.9
Country							
Dominican Republic	1481 (16.2)	41.8	55.9	58.0	26.6	36.0	31.7
Jamaica	1667 (18.2)	48.1	67.9	65.0	31.5	52.4	40.3
Suriname	2126 (23.3)	32.6	49.5	51.4	30.6	51.0	24.0
Trinidad and Tobago	3869 (42.3)	26.4	47.9	43.3	33.8	45.1	16.5
Age in years							
13 or less	2406 (26.4)	24.6	63.4	46.3	30.4	41.0	14.5
14–15	4017 (44.1)	33.3	73.8	53.5	28.0	43.9	26.6
16 or more	2677 (29.4)	52.6	49.8	63.3	28.3	40.8	41.0
Sex							
Female	4816 (50.7)	26.8	36.1	37.5	28.8	35.9	16.2
Male	4221 (49.3)	55.9	69.6	69.1	27.9	45.1	48.5
**Emotional-contextual factors**							
Psychological distress							
Low	5028 (62.3)	36.6	60.4	60.9	24.8	42.9	28.8
Moderate	1788 (19.4)	46.0	59.3	57.9	37.2	45.8	35.7
High	1665 (18.4)	48.4	50.1	54.3	30.8	36.4	34.5
Bullied in past month (3–30 days)	822 (9.1)	50.7	53.9	57.9	33.5	48.6	35.5
In physical fight in past year (≥2 times)	1399 (12.9)	62.8	75.4	63.2	29.8	51.0	54.4
Mostly/always feeling hungry	701 (4.6)	50.5	60.9	64.4	35.4	38.5	40.8
Parental tobacco use	2190 (16.2)	54.0	60.4	57.9	26.7	50.4	41.9
Passive smoking (all past 7 days)	1733 (16.7)	52.7	65.7	68.6	33.6	50.0	44.4
School truancy (3–30 days)	774 (9.5)	65.6	59.1	62.4	36.9	37.3	53.1
**Health risk behaviors**							
Current tobacco use	1222 (13.8)	72.5	74.5	70.8	29.1	41.5	63.6
Current cannabis use	563 (6.4)	79.4	72.7	63.8	36.7	55.0	70.6
Ever drunk	2148 (28.6)	63.5	64.5	66.5	32.3	45.9	54.1
No physical education	2674 (30.7)	49.2	53.4	59.7	30.0	40.0	38.1
Injury in past 12 months (≥2 times)	1492 (15.5)	53.3	57.6	60.2	30.6	46.4	41.4
Soft drink intake (≥2 drinks/day)	3899 (46.8)	45.0	60.1	62.1	28.3	45.6	36.1
**Protective factors**							
Peer support							
Low	3004 (31.2)	46.3	59.3	52.7	33.7	42.6	35.2
Moderate	2701 (29.3)	37.1	60.6	59.5	30.6	47.1	28.4
High	3052 (39.6)	39.2	56.8	63.1	23.6	39.2	31.2
Parental support							
Low	3554 (38.9)	48.2	61.3	58.5	33.4	46.9	37.8
Moderate	2249 (27.5)	38.5	57.9	63.6	24.3	45.2	30.2
High	2526 (33.6)	32.2	53.6	53.9	27.1	33.7	24.4

^a^ Of sexually active.

**Table 4 behavsci-10-00166-t004:** Associations with sexual initiation, early sexual debut, and multiple sexual partners.

Variable	Sexual Initiation	Early Sexual Debut	Multiple Sexual Partners
	AOR (95% CI)	AOR (95% CI)	AOR (95% CI)
Sociodemographic factors			
Country			
Dominican Republic	1 (Reference)	1 (Reference)	1 (Reference)
Jamaica	1.13 (0.72, 1.78)	1.61 (0.87, 2.98)	1.70 (0.90, 3.20)
Suriname	0.66 (0.40, 1.08)	0.55 (0.36, 0.86) **	0.54 (0.27, 1.09)
Trinidad and Tobago	0.47 (0.29, 0.75) **	0.44 (0.24, 0.81) **	0.39 (0.19, 0.80) *
Age in years			
15 or less	1 (Reference)	1 (Reference)	1 (Reference)
16 or more	2.75 (1.66, 4.55) ***	0.92 (0.67, 1.27)	2.59 (1.84, 3.65) ***
Sex			
Female	1 (Reference)	1 (Reference)	1 (Reference)
Male	4.41 (3.62, 5.36) ***	7.26 (5.89, 9.62) ***	7.95 (5.39, 11.84) ***
**Emotional-contextual factors**			
Psychological distress			
Low	1 (Reference)	1 (Reference)	1 (Reference)
Moderate	1.49 (1.15, 1.93) **	1.45 (1.07, 1.96) *	1.33 (0.83, 2.14)
High	1.56 (1.12, 2.16) **	1.62 (1.04, 2.54) *	1.61 (1.13, 2.30) **
Bullied in past month (3–30 days)	0.97 (0.67, 1.42)	0.86 (0.51, 1.54)	0.75 (0.44, 1.25)
In physical fight in past year (≥2 times)	1.77 (1.25, 2.49) ***	1.83 (1.14, 2.95) *	1.67 (1.05, 2.64) *
Mostly/always feeling hungry	0.90 (0.50, 1.61)	0.56 (0.33, 1.00)	1.07 (0.36, 3.14)
Parental tobacco use	1.54 (1.08, 2.20) *	1.09 (0.74, 1.64)	1.38 (0.97, 1.95)
Passive smoking (all past 7 days)	1.08 (0.74, 1.57)	1.14 (0.76, 1.72)	1.34 (0.83, 2.18)
School truancy (3–30 days)	2.31 (1.58, 3.36) ***	1.59 (0.84, 3.01)	1.61 (0.86, 3.01)
**Health risk behaviors**			
Current tobacco use	1.28 (0.84, 1.96)	2.27 (1.42, 3.61) ***	2.14 (1.37, 3.34) ***
Current cannabis use	2.66 (1.61, 4.40) ***	2.10 (1.05, 4.17) *	1.20 (0.60, 2.45)
Ever drunk	2.65 (1.90, 3.70) ***	2.63 (1.88, 3.68) ***	2.66 (1.98, 3.57) ***
No physical education	1.23 (0.89, 1.70)	1.12 (0.71, 1.74)	1.40 (0.91, 2.15)
Injury in past 12 months (≥2 times)	1.14 (0.79, 1.63)	0.82 (0.56, 1.19)	1.06 (0.75, 1.48)
Soft drink intake (≥2 drinks/day)	1.28 (1.08, 1.57)*	1.28 (0.96, 1.70)	1.52 (1.12, 2.06) **
**Protective factors**			
Peer support			
Low	1 (Reference)	1 (Reference)	1 (Reference)
Moderate	0.73 (0.50, 1.07)	0.98 (0.66, 1.47)	0.96 (0.60, 1.54)
High	0.82 (0.58, 1.16)	1.02 (0.70, 1.46)	1.45 (1.08, 1.95) *
Parental support			
Low	1 (Reference)	1 (Reference)	1 (Reference)
Moderate	0.90 (0.71, 1.15)	1.02 (0.62, 1.70)	1.15 (0.79, 1.66)
High	1.01 (0.75, 1.36)	1.00 (0.69, 1.45)	1.12 (0.77, 1.65)

AOR = Adjusted Odds Ratio; *** *p* < 0.001; ** *p* < 0.01; * *p* < 0.05.

**Table 5 behavsci-10-00166-t005:** Associations with non-condom and non-birth control use and multiple sexual risk behaviors.

Variable	Non-Condom Use	Non-Birth Control Use	Multiple Sexual Risk Behaviors
	AOR (95% CI)	AOR (95% CI)	AOR (95% CI)
Sociodemographic factors			
Country			
Dominican Republic	1 (Reference)	1 (Reference)	1 (Reference)
Jamaica	1.05 (0.61, 1.80)	1.40 (0.76, 2.56)	1.54 (0.93, 2.57)
Suriname	0.84 (0.48, 1.49)	0.68 (0.42, 1.10)	0.68 (0.41, 1.13)
Trinidad and Tobago	0.82 (0.46, 1.44)	0.63 (0.34, 1.16)	0.44 (0.24, 0.79) **
Age in years			
15 or less	1 (Reference)	1 (Reference)	1 (Reference)
16 or more	1.73 (1.31, 2.28) ***	1.74 (1.14, 2.65) *	2.89 (1.68, 4.98) ***
Sex			
Female	1 (Reference)	1 (Reference)	1 (Reference)
Male	2.89 (1.91, 4.40) ***	3.84 (2.45, 6.02) ***	7.16 (5.61, 9.15) ***
**Emotional-contextual factors**			
Psychological distress			
Low	1 (Reference)	1 (Reference)	1 (Reference)
Moderate	1.83 (1.12, 2.99) *	1.18 (0.86, 1.63)	1.55 (1.10, 2.18) *
High	2.05 (1.25, 3.35) **	1.18 (0.76, 1.82)	1.82 (1.28, 2.57) ***
Bullied in past month (3–30 days)	1.15 (0.72, 1.85)	1.15 (0.66, 2.02)	0.67 (0.42, 1.08)
In physical fight in past year (≥2 times)	1.13 (0.80, 1.60)	1.73 (1.24, 2.42) **	1.90 (1.22, 2.95) **
Mostly/always feeling hungry	1.20 (0.52, 2.75)	0.84 (0.29, 2.42)	0.73 (0.35, 1.55)
Parental tobacco use	0.90 (0.62, 1.31)	2.12 (1.47, 3.06) ***	1.18 (0.89, 1.56)
Passive smoking (all past 7 days)	1.45 (0.98, 2.16)	1.18 (0.66, 2.11)	1.06 (0.73, 1.52)
School truancy (3–30 days)	1.70 (0.96, 3.01)	1.05 (0.71, 1.56)	2.23 (1.48, 3.36) ***
**Health risk behaviors**			
Current tobacco use	1.00 (0.83, 1.58)	0.87 (0.53, 1.44)	1.64 (1.05, 2.56) *
Current cannabis use	1.66 (1.47, 2.81) ***	2.16 (1.26, 3.59) **	2.57 (1.51, 4.38) ***
Ever drunk	2.03 (0.79, 1.81)	1.93 (1.28, 2.90) **	3.20 (2.42, 4.23) ***
No physical education	1.20 (0.71, 1.61)	1.19 (0.76, 1.85)	1.19 (0.79, 1.78)
Injury in past 12 months (≥2 times)	1.34 (0.72, 2.51)	1.28 (0.88, 1.85)	1.07 (0.67, 1.69)
Soft drink intake (≥2 drinks/day)	1.07 (0.71, 1.61)	1.58 (1.07, 2.32) *	1.47 (1.09, 1.98) *
**Protective factors**			
Peer support			
Low	1 (Reference)	1 (Reference)	1 (Reference)
Moderate	0.68 (0.42, 1.09)	0.96 (0.64, 1.45)	0.79 (0.51, 1.21)
High	0.68 (0.40, 1.17)	1.03 (0.70, 1.53)	0.97 (0.67, 1.42)
Parental support			
Low	1 (Reference)	1 (Reference)	1 (Reference)
Moderate	0.61 (0.42, 0.89) **	0.84 (0.61, 1.15)	1.13 (0.80, 1.60)
High	1.17 (0.61, 2.25)	0.74 (0.50, 1.10)	1.25 (0.86, 1.81)

AOR = Adjusted Odds Ratio; *** *p* < 0.001, ** *p* < 0.01, * *p* < 0.05.

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
