# Peer review of "Prevalence and Correlates of Sexual Risk Behavior among School-Going Adolescents in Four Caribbean Countries"

_behavsci, 2020, doi:10.3390/bs10110166_

Round 1

Reviewer 1 Report

Overall this is a very good topic, important and timely, and I recommend publishing with incorporation of the major edits listed below. 

Introduction:

- Excellent framing of the problem around sexual initiation that often occurs in adolescence and the risk taking associated there

- Please clarify what you mean when you include the average fertility rate in the Caribbean and then also include the Dominican Republic and Jamaica

- When you refer to the Caribbean in the introduction, are you referring to all Caribbean countries or the four that are included in this study?

- Would be helpful to have some brief context of each of the four countries and their demographics, school system where students were surveyed, etc.

- In the second to last paragraph of the introduction, I recommend including the years of when some of these studies were published. It would help to contextualize your work within some of this other literature.

Methods:

- Why were these four countries chosen? And why 2016 for the Dominican Republic and Suriname but 2017 for Trinidad and Tobago?

- Throughout the methods section, why are some sentences and phrases in quotations?

- Can you clarify the age ranges for each of these groups of students from the four countries, for ex is Grades 1-3 and other in Suriname roughly equivalent to the ages that students in grades 7th-12th in Jamaica would be?

- Was your questionnaire based off of the citation listed? If so, I recommend stating that. How was it adapted to your study population? Why did you ask about cannabis use but not other drug use?

- How was confidentiality achieved? Was there support in place in case completing the survey was triggering psychologically for participants?

- How was the survey administered?

- Please clearly define school truancy and passive smoking

Results:

- Recommend rephrasing the second sentence to read: “Among those who endorsed ever being sexually active…”

- What types of birth control were young people using?

- Very interesting that ”high peer support” increased the odds for multiple sexual partners. This is interesting and potentially counterintuitive — glad that it is addressed in the discussion and I think you could delve into it even more and what that means for interventions.

Discussion:

- Overall, I appreciate the organization of the discussion section around salient themes. I recommend including an introductory paragraph that provides a road map for the points that you will make throughout the discussion. I also recommend reading through this section to ensure it flows well. Finally a few minor errors were noted — for example, the first sentence of the paragraph before “Limitations of the study” should read “The study showed that attending school was protective…”

- Please reword the current first sentence of the discussion — this should be broken up in multiple sentences and please explain the comparison to other regions of the world.

- The last sentence of the first paragraph — please contextualize the specification of Kingston and St Andrew

- Why do you think there is increased condom use being found in your results? Also, non-birth control use seems to be very elevated which contradicts this point as there appears to be a discrepancy between condom use vs not using birth control (though condoms can often be considered a form of birth control) — this would be worth exploring.

- Please rephrase the first sentence of the fifth paragraph

- I really like the interventions suggested about attending school in the sixth paragraph. Are there ways to propose other interventions in some of the other sections of the discussion?

Conclusion:

- Please re-word the first sentence of the conclusion as it is difficult to understand.

- I recommend clarifying in the introduction whether you think that these results would apply to the Carribean as a whole in addition to the four countries studied

Author Response

Overall this is a very good topic, important and timely, and I recommend publishing with incorporation of the major edits listed below.
Introduction:
- Excellent framing of the problem around sexual initiation that often occurs in adolescence and the risk taking associated there
- Please clarify what you mean when you include the average fertility rate in the Caribbean and then also include the Dominican Republic and Jamaica
Response: clarified as below
The adolescent fertility rate (defined as the number of births per 1,000 women age 15 to 19 years)
- When you refer to the Caribbean in the introduction, are you referring to all Caribbean countries or the four that are included in this study?
Response: it is specified, if “Caribbean” then it is all countries, if specific countries, they are specified
- Would be helpful to have some brief context of each of the four countries and their demographics, school system where students were surveyed, etc.
Response: below is added under the method section
Study context characteristics of the four study countries are described in Table 1.
Table 1: Study context characteristics
Variable Dominican Republic Jamaica Suriname Trinidad and Tobago
Population 10,499,704 2,808,570 609,569 1,208,789
Urban population 82.5% 56.3% 66.1% 53.2%
Major ethnic groups mixed 70.4% (mestizo/indio 58%, mulatto 12.4%), black 15.8%, white black 92.1%, mixed 6.1%, East Hindustani 27.4%, "Maroon" 21.7%, Creole 15.7%, Javanese 13.7%, mixed East Indian 35.4%, African descent 34.2%, mixed - other 15.3%
Secondary school gross enrolment ratio, female 85% 86% 89% 88%
Secondary school gross enrolment ratio, male 78% 85% 67% 83%
Study sample: Mean age (SD), age range 14.9 (1.5), 11-18 years 15.0 (1.3), 11-18 years 14.8 (1.7), 11-18 years 14.8 (1.7), 11-18 years
School grade
1st Form 426 (29.4%) 41 (2.5%) 642 (31.0%) 896 (23.3%)
2nd Form 394 (27.2%) 412 (25.0%) 678 (32.7%) 667 (17.4%
3rd Form 317 (21.9%) 557 (33.8%) 691 (33.3%) 851 (22.1%)
4th Form 216 (14.9%) 388 (23.5%) 61 (2.9%) 767 (20.0%)
5th Form 94 (6.5%) 208 (12.6%) 0 (0.0%) 462 (12.0%)
6th Form 0 (0.0%) 44 (2.7%) 0 (0.0) 200 (5.2%)

- In the second to last paragraph of the introduction, I recommend including the years of when some of these studies were published. It would help to contextualize your work within some of this other literature.
Response: below is added
In several local studies among adolescents in Jamaica, high sexual risk behaviour has been reported, e.g., in a subsample of adolescents (15-19 years) in 2008-2009, 32% of females and 54% of males had sexual initiation, and among sexually active girls 12% and boys 52% had multiple sexual partners [6], in an adolescent school survey (13-18 years) in Kingston and St Andrew in 2012 showed that 58.6% reported sexual initiation and 31.8% had multiple sexual partners [7], and among sexually active youth (15-24 years) in Jamaica in 2012, 58.8% reported inconsistent condom use, and 44.5% had multiple sexual partners [8].

Methods:
- Why were these four countries chosen? And why 2016 for the Dominican Republic and Suriname but 2017 for Trinidad and Tobago?
Response: below is added
These countries were selected on the basis of all countries in the Caribbean where recent GSHS data were available.
- Throughout the methods section, why are some sentences and phrases in quotations?
Response: Corrected
- Can you clarify the age ranges for each of these groups of students from the four countries, for ex is Grades 1-3 and other in Suriname roughly equivalent to the ages that students in grades 7th-12th in Jamaica would be?
Response: this is clarified in below table
Table 1: Study context characteristics
Variable Dominican Republic Jamaica Suriname Trinidad and Tobago
Population 10,499,704 2,808,570 609,569 1,208,789
Urban population 82.5% 56.3% 66.1% 53.2%
Major ethnic groups mixed 70.4% (mestizo/indio 58%, mulatto 12.4%), black 15.8%, white black 92.1%, mixed 6.1%, East Hindustani 27.4%, "Maroon" 21.7%, Creole 15.7%, Javanese 13.7%, mixed East Indian 35.4%, African descent 34.2%, mixed - other 15.3%
Secondary school gross enrolment ratio, female 85% 86% 89% 88%
Secondary school gross enrolment ratio, male 78% 85% 67% 83%
Study sample: Mean age (SD), age range 14.9 (1.5), 11-18 years 15.0 (1.3), 11-18 years 14.8 (1.7), 11-18 years 14.8 (1.7), 11-18 years
School grade
1st Form 426 (29.4%) 41 (2.5%) 642 (31.0%) 896 (23.3%)
2nd Form 394 (27.2%) 412 (25.0%) 678 (32.7%) 667 (17.4%
3rd Form 317 (21.9%) 557 (33.8%) 691 (33.3%) 851 (22.1%)
4th Form 216 (14.9%) 388 (23.5%) 61 (2.9%) 767 (20.0%)
5th Form 94 (6.5%) 208 (12.6%) 0 (0.0%) 462 (12.0%)
6th Form 0 (0.0%) 44 (2.7%) 0 (0.0) 200 (5.2%)

- Was your questionnaire based off of the citation listed? If so, I recommend stating that. How was it adapted to your study population? Why did you ask about cannabis use but not other drug use?
Response, as below
The GSHS questionnaire used is shown in Table 2 [15].
We analysed available data
- How was confidentiality achieved? Was there support in place in case completing the survey was triggering psychologically for participants? - How was the survey administered?
Response: below is added
Data collection was done with a self-administered multi-choice format questionnaire translated into the country language under the supervision of trained external survey administrators [15]. Students recorded their responses on computer scannable sheets, and student privacy was maintained through anonymous and voluntary participation [15]
- Please clearly define school truancy and passive smoking
Response: this is defined in the questionnaire table, as below
Passive smoking “During the past 7 days, on how many days have people smoked in your presence?” “1=0 days to 5=all 7 days (coded 1-4=0 and 5=1)”
School truancy “During the past 30 days, on how many days did you miss classes or school without permission?” “1=0 days to 10 or more days (coded 1-2=0 and 3-5=1)”

Results:
- Recommend rephrasing the second sentence to read: “Among those who endorsed ever being sexually active…”
Response: Corrected
- What types of birth control were young people using?
Response: this is described the questionnaire section, as below
Birth control use “The last time you had sexual intercourse, did you or your partner use any method of birth control, such as withdrawal, rhythm (safe time), birth control pills, or any other method to prevent pregnancy?” “I have never had sexual intercourse, Yes, No, I do not know”

- Very interesting that ”high peer support” increased the odds for multiple sexual partners. This is interesting and potentially counterintuitive — glad that it is addressed in the discussion and I think you could delve into it even more and what that means for interventions.
Response: more is added
Discussion:
- Overall, I appreciate the organization of the discussion section around salient themes. I recommend including an introductory paragraph that provides a road map for the points that you will make throughout the discussion. I also recommend reading through this section to ensure it flows well. Finally a few minor errors were noted — for example, the first sentence of the paragraph before “Limitations of the study” should read “The study showed that attending school was protective…”
Response: corrected
- Please reword the current first sentence of the discussion — this should be broken up in multiple sentences and please explain the comparison to other regions of the world.
Response: corrected
- The last sentence of the first paragraph — please contextualize the specification of Kingston and St Andrew
Response: it already says in a study in Jamaica
- Why do you think there is increased condom use being found in your results? Also, non-birth control use seems to be very elevated which contradicts this point as there appears to be a discrepancy between condom use vs not using birth control (though condoms can often be considered a form of birth control) — this would be worth exploring.
Reponse: below is added
It is possible that increased HIV prevention campaigns led to a high proportion of condom use among the studied adolescents.

- Please rephrase the first sentence of the fifth paragraph
Response: corrected
- I really like the interventions suggested about attending school in the sixth paragraph. Are there ways to propose other interventions in some of the other sections of the discussion?
Conclusion:
- Please re-word the first sentence of the conclusion as it is difficult to understand.
Response: corrected
- I recommend clarifying in the introduction whether you think that these results would apply to the Carribean as a whole in addition to the four countries studied
Response: some data refer to the Caribbean as a whole and our data to four countries, this is always specified

Reviewer 2 Report

This article provides an interesting cross-national comparison of a large data set in regards factors influencing sexual risk behaviors among adolescents. With an abundance of variables examined, I am interested to know more about what the authors conclude from this information in regards to implications for interventions and health promotion. Examples of why and what types of interventions are important.

There are some major formatting issues, and more clarity is needed in the methods/measures section. I have outlined line by line details below.

Lines 14-15

Abstract: Issue with quotations

Lines 34-40

Language changes to past tense (should either reference a specific study, or leave as present tense).

Lines 34-44

There are variations in font color and style

Line 64

Can you define “safer sex”?

Methods:

Is this a secondary analysis? I see references and quotations from a previously publicized article. Please be clear. The location of where the GSHS health data should also be referenced here (not just reference 15, the previously publicized article).

2.1 Sample and procedure:

What are the ages of the participants? Did they differ based on country?

Measures:

Again, unclear why the authors are being referenced (reference #15), as opposed to the WHO GSHS health data source. My understanding is that the measures came from the WHO survey. It would be good to have subheadings in measures, to define which specific variables the authors are looking at i.e. sexual risk behaviors, psychological distress, parental or guardian support. Looking at Table 1, it appears there are many more categories. Is there anything that can be said about the reliability and validity of this survey, or why the categories were chosen?

Lines 89-90

Unclear on the measure of 0, single, multiple. Is this referring to friends, measures of loneliness, anxiety, suicidal ideation, or suicide attempt, or psychological distress in general? Please be clear.

Line 100

Unclear on ‘measure of four sexual risk behaviors (≥2 risk behaviors).’ Are you looking at how many participants have more than four sexual risk behaviors, or more than two?

Table 2, table 3, table 4:

There are formatting issues throughout, with the numbers not lining up with the study variables

Line 153

Unclear why other continents/countries are listed if referencing Caribbean data (sentence format unclear).

Lines 153-172

There are variations in font color and style

Line 175

Remove _

Line 178

‘highest in the Caribbean’…when compared to Europe, Israel, and Canada? Unclear when comparing to individual countries, combined with continents; is that an accurate comparison? Please explain.

Line 200

Reference bracket typo

Line 208

Attending school?

Author Response

This article provides an interesting cross-national comparison of a large data set in regards factors influencing sexual risk behaviors among adolescents. With an abundance of variables examined, I am interested to know more about what the authors conclude from this information in regards to implications for interventions and health promotion. Examples of why and what types of interventions are important.

There are some major formatting issues, and more clarity is needed in the methods/measures section. I have outlined line by line details below.

Lines 14-15
Abstract: Issue with quotations
Response: Corrected
Lines 34-40
Language changes to past tense (should either reference a specific study, or leave as present tense).
Response: Corrected
Lines 34-44
There are variations in font color and style
Response: Corrected
Line 64
Can you define “safer sex”?
According to UNAIDS, “Safer sex strategies include postponing sexual debut, non-penetrative sex, correct and consistent use of male or female condoms, and reducing the number of sexual partners” [15] (p.11).
Methods:
Is this a secondary analysis? I see references and quotations from a previously publicized article. Please be clear. The location of where the GSHS health data should also be referenced here (not just reference 15, the previously publicized article).
Response: corrected
2.1 Sample and procedure:
What are the ages of the participants? Did they differ based on country?
Response: an extra Table 1 has been added showing the ages of the participants by country
Measures:
Again, unclear why the authors are being referenced (reference #15), as opposed to the WHO GSHS health data source. My understanding is that the measures came from the WHO survey.
Response: corrected

It would be good to have subheadings in measures, to define which specific variables the authors are looking at i.e. sexual risk behaviors, psychological distress, parental or guardian support. Looking at Table 1, it appears there are many more categories.
Response: not sure why they should all be repeated in the measures section, but as you wish, this is added
Is there anything that can be said about the reliability and validity of this survey, or why the categories were chosen?
Response: below is added
The GSHS measure draws on content from the CDC Youth Risk Behavior Survey for which test- and retest reliability has been established (Brener, Collins, Kann, Warren, & Williams, 1995). In a study examining the test-retest reliability of the GSHS measure among Fijian girls found an “average agreement between test and retest was 77%, and average Cohen's kappa was 0.47.” (Becker et al., 2010, p.181).
Co-variates were included based on previous literature review [11-14].
Lines 89-90
Unclear on the measure of 0, single, multiple. Is this referring to friends, measures of loneliness, anxiety, suicidal ideation, or suicide attempt, or psychological distress in general? Please be clear.
Response: corrected
Line 100
Unclear on ‘measure of four sexual risk behaviors (≥2 risk behaviors).’ Are you looking at how many participants have more than four sexual risk behaviors, or more than two?
Response: corrected

Table 2, table 3, table 4:
There are formatting issues throughout, with the numbers not lining up with the study variables
Response: corrected

Line 153
Unclear why other continents/countries are listed if referencing Caribbean data (sentence format unclear).
Response: This should be clear comparing our results with other regions
Lines 153-172
There are variations in font color and style
Response: corrected
Line 175
Remove _
Response: corrected
Line 178
‘highest in the Caribbean’…when compared to Europe, Israel, and Canada? Unclear when comparing to individual countries, combined with continents; is that an accurate comparison? Please explain.
Response: The results of the HSBC study include a number of European countries and two non-European countries (Israel and Canada) [not individual countries]
Line 200
Reference bracket typo
Response: corrected
Line 208
Attending school?
Response: corrected

Reviewer 3 Report

The work presented is very interesting. Contemplating the risks linked to adolescent sexual practices has enormous clinical and educational implications.

The authors use a sample of enormous dimensions and the methodology is robust.

However, it is surprising that at no time do they contemplate aspects linked to the sexual diversity of the subjects in the sample. Do heterosexual boys start having sex at the same time as homosexual boys? Do they have the same type of risky practice?

This reflection is fundamental in the introduction as well as in the analysis and discussion.

Author Response

The work presented is very interesting. Contemplating the risks linked to adolescent sexual practices has enormous clinical and educational implications.
The authors use a sample of enormous dimensions and the methodology is robust.
However, it is surprising that at no time do they contemplate aspects linked to the sexual diversity of the subjects in the sample. Do heterosexual boys start having sex at the same time as homosexual boys? Do they have the same type of risky practice?
This reflection is fundamental in the introduction as well as in the analysis and discussion.
Response: Sexual orientation was unfortunately not assessed in this study, and can therefore also not be discussed

Reviewer 4 Report

the manuscript presents a very interesting study. the data seem to support the authors' hypotheses. However, I believe that some changes can be made: 1. the introduction appears to be a bit poor, while in the literature there are many studies on the risk factors of sexual behavior; 2. there is a long tradition of the studies conducted by Pinel Institut which could also find application in this study; 3. the investigation tool should be better explained: the number of items, in which language they were presented and whether all subjects used the same version or translated into the local language 4. the validity of the instrument. 5. the authors indicated no risk factors. it would be appropriate to specify from which evidence or studies have deducted the variables considered. 6. Study participants come from different countries. how did the authors manage cultural factors? 7. How do protective factors interact with respect to gender and age? the authors could also implement the analysis models. 8. a more detailed description of the sample could help in reading the results: the average age for each country of both males and females. at what time were all the protocols collected? 9. Are the variables evenly distributed among all the countries considered, in particular age, gender and distribution of sexual behavior?

Author Response

the manuscript presents a very interesting study. the data seem to support the authors' hypotheses. However, I believe that some changes can be made: 1. the introduction appears to be a bit poor, while in the literature there are many studies on the risk factors of sexual behavior;
Response: the many studies are introduced in the discussion
2. there is a long tradition of the studies conducted by Pinel Institut which could also find application in this study;
Response: could you explain this further, this is unclear
3. the investigation tool should be better explained: the number of items,
Response: below has additions
The GSHS questionnaire used is shown in Table 2 [15]. The GSHS measure draws on content from the CDC Youth Risk Behavior Survey for which test- and retest reliability has been established (Brener, Collins, Kann, Warren, & Williams, 1995). In a study examining the test-retest reliability of the GSHS measure among Fijian girls found an “average agreement between test and retest was 77%, and average Cohen's kappa was 0.47.” (Becker et al., 2010, p.181).
Sexual risk behaviour was assessed with questions on ever having had sexual intercourse, age of sexual debut, number of people having had sexual intercourse within a lifetime, condom use at last sexual intercourse, and birth control use at last sexual intercourse. Individual sexual risk behaviours were defined as ever having had sex, early sexual debut (≤14 years), having had two or more sexual partners in a lifetime, non-condom use at last sex and non-birth control use at last sex. A composite sexual risk behaviour measure included having had sex, early sexual debut (≤14 years), having had two or more sexual partners in a lifetime, and noncondom use at last sex; non-birth control use was excluded due to overlap with noncondom use at last sex.
Emotional-contextual factors bullied, in physical fights, hunger, parental tobacco use, passive smoking, school truancy, and psychological distress variables (loneliness, anxiety, no close friends, suicidal ideation, and suicide attempt) that were summed and grouped into 0=0, 1=1 single, and 2-5=2 multiple psychological distress [15].
Health risk behaviours included current tobacco use, current cannabis use, lifetime drunkenness, attendance of physical education, injury, and soft drink consumption.
Protective factor items included peer support and four parental support items (connectedness, supervision, bonding, and respect for privacy). The latter were summed and classified into three groups, 0-1 low, 2 medium, and 3-4 high support [16].

in which language they were presented and whether all subjects used the same version or translated into the local language 4.
Response: this is added, e.g., below
At the first stage, schools were selected with probability proportional to enrolment size, and at the second stage, classes were randomly selected and all students in selected classes were eligible to participate [15]. Data collection was done with a self-administered multi-choice format questionnaire translated into the country language under the supervision of trained external survey administrators [15]. Students recorded their responses on computer scannable sheets, and student privacy was maintained through anonymous and voluntary participation [15].
the validity of the instrument.
Response: below is added
The GSHS measure draws on content from the CDC Youth Risk Behavior Survey for which test- and retest reliability has been established (Brener, Collins, Kann, Warren, & Williams, 1995). In a study examining the test-retest reliability of the GSHS measure among Fijian girls found an “average agreement between test and retest was 77%, and average Cohen's kappa was 0.47.” (Becker et al., 2010, p.181).

5. the authors indicated no risk factors. it would be appropriate to specify from which evidence or studies have deducted the variables considered.
Response: below is added
Co-variates were included based on previous literature review [11-14]
6. Study participants come from different countries. how did the authors manage cultural factors?
Response: More cultural information is added in a new Table on study context. Country differences were assessed and discussed
7. How do protective factors interact with respect to gender and age?
Response: we did not find any interaction effects
the authors could also implement the analysis models.
Response: unclear what this means
8. a more detailed description of the sample could help in reading the results: the average age for each country of both males and females. at what time were all the protocols collected?
Response: More to this effect is added
9. Are the variables evenly distributed among all the countries considered, in particular age, gender and distribution of sexual behavior?
Response: in an extra table age and school form distribution by country is added. Sexual behaviour country differences are described in Table 3-5

Round 2

Reviewer 2 Report

Authors have made suggested changes.

Please proofread for final format and spellcheck. 

Author Response

Please proofread for final format and spellcheck.

corrected

Reviewer 3 Report

The work has been substantially improved.

Author Response

English language and style are fine/minor spell check required, corrected

Reviewer 4 Report

The authors made only some of the required changes to the manuscript.

the author
deal with the incidence of certain factors that can lead to sexual.
misconduct.

It is the opinion of this reviewer that the study of previous studies dealing with this topic should be increased   The authors report that they added the covariates in the analyzes but I cannot identify them in the changes made to the manuscript The authors report that to respond to my previous indications they could carry out further analyzes. Claim that there is no effect between protective and risk factors. This is certainly a limit to the study that should be discussed. They should try to perform further analysis the tools used are poor in description in the manuscript. Authors could add information on the validity of the tools and possibly also with respect to their study by presenting the coefficients of their sample.      

Author Response

The authors made only some of the required changes to the manuscript.

the author deal with the incidence of certain factors that can lead to sexual.
misconduct.

It is the opinion of this reviewer that the study of previous studies dealing with this topic should be increased
Response: We said the previous studies appear in the discussion, as below
The results show a high proportion of sexual initiation (41.4%) among school-going adolescents in four Caribbean countries, which seems higher than 2013 estimates for the Caribbean (20% among girls and 40% among boys) [2], in a community survey among adolescents in seven African countries (25.9%) [25], among school adolescents in 30 countries in Europe, Israel and Canada (27%) [26], and in a 10 European countries study among adolescents (18.8%) [27]. There was some country variation in the prevalence of sexual initiation, with Jamaica being the highest (48.1%) and Trinidad and Tobago the lowest (26.4%). Two previous local studies among adolescents in Jamaica also reported a high proportion of sexual initiation (32% among girls and 54% among boys) [6], and in Kingston and St Andrew (58.6%) [7].
Among sexually active students, 28.4% had not used condoms at the last sexual encounter, which is a significant improvement compared to 2013 or earlier data among adolescents in the Caribbean (38%) [2], Haiti (42.3%) [3] and in seven African countries (46% among girls and 49% among boys) [25], four countries in Southeast Asia (46.9%) [24], but similar to adolescents in Uganda (22.7%) [28]. Moreover, high sexual risk behaviour was found in terms of early sexual debut (≤15 years) (58.8%), multiple sex partners (58.8%), non-birth control use (last sex) (41.9%), and engagement in two or more sexual risk behaviours (31.9%). The proportion of early sexual debut among adolescents in the four Caribbean countries was higher in this study (58.8%) than in a previous study in the Caribbean in 2007-2009 (26.9%) [5], and in seven African countries (<15 years, 21% among girls and 28% among boys) [18], but similar to 2013 or earlier Caribbean research data (56% of girls and 79% of boys) [2]. The proportion of multiple sexual partners (58.8%) among adolescents in the Caribbean in this study was higher than in 2013 research in the Caribbean (31% among boys and 10% among girls) [2], in Kingston and St Andrew in Jamaica (31.8%) [7], in another study in Jamaica (12% among girls and 52% among boys) [6], in Ghana (32.5%) [29], in 15 year-olds in 10 European countries (52.4%) [27], but lower to a local study of 200 adolescents in Haiti (62.2%) [3].
The proportion of non-contraceptive use (41.9%) among adolescents in the Caribbean in this study was higher than in Europe, Israel, and Canada (14%) [26]. Although the prevalence of birth control use (64%) and condom use (73.4%) at last sex among adolescents in Dominican Republic in this study was the highest among the four Caribbean study countries the adolescent fertility rate was the highest in the Caribbean (108.7 per 1,000 adolescent women) [2]. This disparity may be related to our sample selection, being middle school students, rather than out-of school adolescents. Most sexual risk behaviours were the highest in Jamaica and the lowest in Trinidad and Tobago, which concurs with some previous research [6,7]. Overall, it appears that protective sexual intercourse increased but also early sexual debut and multiple sexual partners increased among adolescents in the four Caribbean countries. It is possible that increased HIV prevention campaigns led to a high proportion of condom use among the studied adolescents.

The authors report that they added the covariates in the analyzes but I cannot identify them in the changes made to the manuscript
Response: below is mentioned in the data analysis section
Co-variates were included based on previous literature reviews [11-14,24].

Emotional-contextual factors bullied, in physical fights, hunger, parental tobacco use, passive smoking, school truancy, and psychological distress variables (loneliness, anxiety, no close friends, suicidal ideation, and suicide attempt) that were summed and grouped into 0=0, 1=1 single, and 2-5=2 multiple psychological distress [24].
Health risk behaviours included current tobacco use, current cannabis use, lifetime drunkenness, attendance of physical education, injury, and soft drink consumption.
Protective factor items included peer support and four parental support items (connectedness, supervision, bonding, and respect for privacy). The latter were summed and classified into three groups, 0-1 low, 2 medium, and 3-4 high support [24].

The authors report that to respond to my previous indications they could carry out further analyzes. Claim that there is no effect between protective and risk factors. This is certainly a limit to the study that should be discussed.
Response: Unclear what is meant here. We are not examining the association between protective and risk factors, but between protective, risk factors and sexual risk behaviour
They should try to perform further analysis the tools used are poor in description in the manuscript.
Response: Not clear what further analysis is really needed
Authors could add information on the validity of the tools and possibly also with respect to their study by presenting the coefficients of their sample.
Response: below is described
The GSHS measure draws on content from the CDC Youth Risk Behavior Survey for which test- and retest reliability has been established [22]. In a study examining the test-retest reliability of the GSHS measure among Fijian girls found an “average agreement between test and retest was 77%, and average Cohen's kappa was 0.47.” [23] (p.181).

-we cannot present coefficients of single item measures

Round 3

Reviewer 4 Report

the manuscript may be accept for pubblication